# Rosin Derivatives as a Platform for the Antiviral Drug Design

**DOI:** 10.3390/molecules26133836

**Published:** 2021-06-23

**Authors:** Larisa Popova, Olga Ivanchenko, Evgeniia Pochkaeva, Sergey Klotchenko, Marina Plotnikova, Angelica Tsyrulnikova, Ekaterina Aronova

**Affiliations:** 1Graduate School of Biotechnology and Food Science, Peter the Great St. Petersburg Polytechnic University, Polytechnicheskaya Street 29, 195251 Saint Petersburg, Russia; ivanchenko_ob@spbstu.ru (O.I.); london2295@yandex.ru (A.T.); aronova_eb@spbstu.ru (E.A.); 2Smorodintsev Research Institute of Influenza, Prof. Popov Street 15/17, 197376 Saint Petersburg, Russia; fosfatik@mail.ru (S.K.); biomalinka@mail.ru (M.P.)

**Keywords:** antibacterial activity, antiviral activity, biological activity, cytotoxicity, DAA, fungicidal activity, MTT assay, PASS-online modeling, SWISS-ADME web server

## Abstract

The increased complexity due to the emergence and rapid spread of new viral infections prompts researchers to search for potential antiviral and protective agents for mucous membranes among various natural objects, for example, plant raw materials, their individual components, as well as the products of their chemical modification. Due to their structure, resin acids are valuable raw materials of natural origin to synthesize various bioactive substances. Therefore, the purpose of this study was to confirm the possibility of using resin acid derivatives for the drug design. As a result, we studied the cytotoxicity and biological activity of resin acid derivatives. It was shown that a slight decrease in the viral load in the supernatants was observed upon stimulation of cells (**II**) compared with the control. When using PASS-online modeling (Prediction of Activity Spectra for Substances), the prediction of the biological activity spectrum showed that compound (**I**) is capable of exhibiting antiviral activity against the influenza virus. The use of the SWISS-ADME webserver to reveal the drug-like properties of compounds did not directly indicate the presence of antiviral activity. These results indicate the potential of resin acid derivatives as a starting point for extensive research in the study of biological activity.

## 1. Introduction

The rosin components, in particular, resin acids, have a broad spectrum of biological activity [1,2]. Natural diterpenes and their synthetic analogs are of interest to researchers due to their physiological activity. For example, dehydroabietic acid (DAA) derivatives are characterized by antibacterial action against the Staphylococcus aureus strain and multidrug-resistant strains [3,4,5]. In addition, these derivatives exhibit antimicrobial [6], antitumor activity at the level of commercial 5-fluorouracil [7,8,9], as well as fungicidal activity, potentially valuable in the fight against growing resistance to antimicrobial agents. Nevertheless, the antimicrobial mechanism of resin acids has not been fully studied [10]. The authors [11] found these compounds affect the fatty acids branching in the bacterial cell wall and decrease the electrochemical gradient of protons. This leads to the destruction of the bacterial cell wall and cell agglomeration.

To prove the antitumor activity of resin acids, the authors [12] estimated the effect of abietic acid on cisplatin-resistant nasopharyngeal cancer cells. MTT assay (3-(4,5-dimethylthiazol-2-yl)-2,5-diphenyltetrazolium bromide) was performed to monitor the rate of cell proliferation). It was found that abietic acid has a strong antiproliferative effect against the nasopharynx line of cancerous cells. It was manifested through the induction of apoptosis.

MTT assay was carried out to study the cytotoxicity of two dehydroabietic acid derivatives *N*-(5-dehydroabietyl-1,3,4-thiadiazole)-yl-pyridine-2-carboxamide (DTPC) and di-*N*-(5-dehydroabietyl-1,3,4-thiadiazole)-yl-pyridine-2,6-carboxamide (DDTPC) with 1,3,4-thiadiazole, pyridine and amide moieties [13]. It was shown that the derivatives have selective cytotoxicity, and DTPC exhibited better cytotoxicity. Furthermore, it was established the antiproliferative effect of DTPC toward the A431 cell line was stronger than that of clinically used cisplatin and oxaliplatin. The cytotoxicity of DTPC and DDTPC was closely related to their DNA binding ability.

At the same time, there are data of the DAA rejuvenating effect, which affects the human lifespan through the activation of SIRT1B [14].

Based on the literature data analysis, we have studied the cytotoxicity of compounds (**I**–**IV**, Figure 1), and also the potassium salt of disproportionated rosin (**V**). Using MTT assay, we investigated the antibacterial and fungicidal activity of gum (**VI**), tall oil (**VII**), and disproportionated rosin (**VIII**). It is widely used as an emulsifier in the production of synthetic rubbers and contains dehydroabietic acid (up to 65% DAA)), abietic acid (**IX**, Figure 1), 12-bromo- and 12-sulfodehydroabietic acids (**X** and **I**, Figure 1), potassium salt of 12-sulfodehydroabietic acid (**IV**), abietinol (**XI**, Figure 1).

We studied the antiviral activity of compounds (**I**–**V**), which are intermediates in the synthesis of promising biologically active substances (BAS). We have also carried out a study connected with computer screening of compounds ((**I, III, IV, IX**–**XI**), Appendix A) to identify their pharmacological value using PASS-online modeling. In addition, we used the SWISS-ADME web server to identify the drug-like properties of compounds ((**I, III, IV, IX**–**XI**), Appendix A).

## 2. Materials and Methods

### 2.1. Materials

This paper used gum rosin (**VI**) of «Gum Rosin» trademark (Lesokhimik, Belarus, industry standard 19113-84) softening point 69 °C, acid index 169, ash <0.03%, isomeric resin acids with the general formula C_20_H_30_O_2_ (from 93 to 94%) and neutral unsaponifiable substances (from 6 to 8%), tall oil rosin (**VII**) of «SYLVAROS^®^85» trademark (Kraton Corp., Houston, TX, USA): softening point 63 °C, acid index 168, ester index 175, flash point 225 °C, ash <0.05%, the content of abietic acid 38%, neoabietic acid 3%, palustric acid 8%, pimaric acid 2%, isopimaric acid 3%, dehydroabietic acid 21%, other resin acids, disproportionated rosin (**VIII**), abietic acid **(IX**), abietinol (**XI**), 12-bromo- and 12-sulfodehydroabietic acids (DAA) (**X**, **I**), and potassium salts of disproportionated rosin (**V**), abietic acid (**II**), 12-bromo- and 12-sulfo-DAA (**III**, **IV**).

Disproportionated rosin [15] by its composition is a mixture of dehydro- (65%), dihydro- (21%), and tetrahydro- (5%) abietic acids and other acids (7%) (acid index 158). It was provided for research by the S.V. Lebedev Research Institute of Synthetic Rubber (St. Petersburg).

### 2.2. Synthesis of Compounds and Their Identification

#### 2.2.1. Compound 12-Bromo-DAA

12-Bromo-DAA (**X**) was synthesized by the bromination reaction (H_2_O, Br_2_/KBr, 90 °C, 3 h, 76% yield) 12-sulfo-DAA (**I**) [16]: mp. 197–199 °C; UV spectrum (EtOH), nm (lg ε): 210 (6.11) 276 (shoulder); 1H-NMR (500 MHz, CDCl_3_): δ, 6.90 (s, 1H, C^14^H), 7.35 (s, 1H, C^11^H).

#### 2.2.2. Compound 12-Sulfo-DAA

12-Sulfo-DAA (**I**) was obtained from disproportionated rosin by treating sulfuric acid based on the method [17]: mp. 330 °C; UV spectrum (EtOH), nm (lg ε): 206 (4.53); ^1^H-NMR (500 MHz, CDCl_3_): δ, 6.08 (s, 1H, C^14^H), 7.64 (s, 1H, C^11^H).

#### 2.2.3. Compound Abietic Acid

Abietic acid (**IX**) was isolated from tall oil rosin according to the known method [18] by the isomerization of abietic acid alcoholic solution in the presence of concentrated HCl at 80 °C, for 2 h in an argon atmosphere. After distilling off the solvent, the reaction mixture was washed with water; the isomerized rosin was dissolved in ether, and the ether extract was dried with MgSO_4_. The calculated amount of diethylamine was added to the dried ether extract. The formed crystals of diethyl ammonium abietate were recrystallized twice from petroleum ether. The purified salt was decomposed with acetic acid. The obtained abietic acid was recrystallized once from ethanol, mp. 171–172 °C, [α]D-105°. IR (film), cm^−1^: 3425 (ν*_OH_*), 3065–2535 (ν*_C_*_-*H*_), 1695, 1630 (ν*_C_*_=*O*_). UV (EtOH), nm (lg ε): 242 (4.02), ^1^H-NMR (500 MHz, CDCl_3_): δ, 0.86 (3H, *C*^20^*H*_3_), 1.01 and 1.02 (3H, *C*^16,17^*H*_3_), 1.26 (3H, *C*^19^*H*_3_), 2.24 (m, C^15^H, *J* = 6.89 Hz), 5.36 (s, 1H, C^7^H), 5.78 (s, 1H, C^14^H). Iodine index, g I2/100 g-173.5, acid index, g KOH/100 g-183.

#### 2.2.4. Compound Abietinol

Abietinol (**XI**) was synthesized as described in [19] by the reduction of abietic acid (AA) (**IX**) with lithium aluminum hydride in THF. t. mp. 85.5–87 °C, IR, ν max (KBr) cm^−1^: 3422, 1459, 1384, 1041. UV (EtOH), nm (lg ε): 240–242 (4.1), [α]D_22_-130°, ^1^H-NMR (500 MHz, CDCl_3_): δ, 5.40 (s, 1H, C^7^H), 5.77 (s, 1H, C^14^H).

#### 2.2.5. Compounds Potassium Salts of AA and DAA Derivatives

The general procedure included the following: potassium salts of disproportionated rosin (**V**), abietic rosin (**II**), 12-bromo- (**III**), and 12-sulfo- (**IV**) DAA were synthesized by the interaction of equimolar amounts of an aqueous solution of potash (K_2_CO_3_) with the corresponding DAA derivatives at moderate heating until complete dissolution. The hydrogen index values for these salts solutions: pH = 11 (**II, III,** and **V**), pH = 4 (**IV**).

### 2.3. Cell Cultures and Strains

The test strains of *Escherichia coli*, *Bacillus subtilis*, and *Candida tropicalis* to study antibacterial and fungicidal activity were provided from the collection of cell cultures of the Graduate School of Biotechnology and Food Science (SPbPU). The A549 continuing human cell line (lung carcinoma) was obtained from the American Type Culture Collection (ATCC, CCL-185), lot number 60150896.

### 2.4. Cytotoxicity

The cytotoxicity of compounds (**I**–**V**) was assessed on A549 cell culture using a colorimetric MTT assay. A549 cells were incubated for 24 h at 37 °C, 5% CO_2_ with solutions of the studied compounds at various concentrations. Next, the cell monolayer was washed, and MTT solution was added in a volume of 0.1 mL at a concentration of 0.5 μg/mL. After 1 h of MTT contact with the cells, the wells were washed, and the formed formazan crystals were dissolved in 0.1 mL of DMSO. The results were recorded at 560 nm on a CLARIOstar Plus plate photometer (BMG LABTECH, Ortenberg, Germany). The IC50 was calculated using GraphPad Prism 6.0.

### 2.5. Antibacterial and Fungicidal Activity

The determination of compounds (**I, IV, VI**–**XI**) effect concerning microorganisms was carried out using test strains of *Escherichia coli*, *Bacillus subtilis*, and *Candida tropicalis*. They are not only natural inhabitants of natural biocenoses but also a part of human microbiocenosis. Used microorganisms of bacteria were selected as representatives of the Gram-positive and Gram-negative human microbiota.

The antibacterial activity of compounds (**I, IV, VI**–**XI**) was studied by the disco-diffuse method [20,21]. The strain culture was transferred sterile from the mown agar into test tubes with 5 mL of nutrient medium to obtain a “night culture” 12–15 h before the experiment. Beef-extract agar (BEA) was used as a nutrient medium for the development of bacterial cells, and Sabouraud’s medium was used for yeast. Bacterial cultures were incubated in a thermostat at 37 °C, and yeast cultures at 32 °C. The melted BEA was poured sterile into Petri dishes (diameter 90 mm) and evenly distributed over the bottom of the dish, covered with a lid, and left on the table until it solidified completely. Next, the dishes with BEA were inoculated with a “lawn” by introducing a strain culture (suspension titer −1–2 × 10^8^ CFU/mL) in an amount of 500 µL and evenly distributing it with a sterile spatula. The dishes were left for 60–90 min to dry the culture.

Compounds (**I**, **IV**, **VI**–**XI**) were previously dissolved, and the sequential dilutions were applied to disks. Dimethyl sulfoxide (DMSO) was used as a solvent for the DAA. Water was used as a solvent for their potassium salts. A sterile paper disk (diameter 16 mm) was placed in the middle of the dish on the lawn surface. This disk was impregnated with a solution of the test compound amounting to 100 μL and incubated in the dark for 24 h at 37 °C (*Escherichia coli* and *Bacillus subtilis*) and 48 h at 32 °C (*Candida tropicalis*).

The solution diffuses, forming a concentration gradient of the test compound around the disc. After the incubation period, the inhibition zone or stimulation of microbial growth around the disk was measured in the experimental and control samples. A conclusion was made about the degree (bactericidal) toxicity of compounds. Solvent-treated disks (DMSO) were used as controls. The experiments were performed three times.

### 2.6. Antiviral Activity

The influenza virus was obtained from the Virus and Cell Culture Collection of the Smorodintsev Research Institute of Influenza (St. Petersburg, Russia). The A/California/07/09 ((A)H1N1pdm09) viral strain was used for infection. The assessment of the antiviral action of compounds (**I**–**V**) was carried out on the A549 cell culture according to the therapeutic scheme. First, the cells were infected with the influenza virus in various doses, then 2.5 h later, they were stimulated with the studied compounds. Concentrations two or more times less than 50% of the toxic dose were taken as the working concentration of the investigated compounds, except for compounds (**I**) and (**II**).

Then, 24 h after stimulation of the cells infected with the influenza A virus (HAV) with the studied compounds, the number of viral particles was estimated by the ELISA method (by the structural protein of the influenza NP virus) in the supernatants obtained from cells, as well as the intracellular (in the cell) content of the viral protein NP (nucleoprotein).

### 2.7. Enzyme-Linked Immunosorbent Assay

ELISA was performed using 96-well Nunc MaxiSorp plates (Thermo Scientific Nalgene, Rochester, NY, USA). Monoclonal antibodies (1 μg/mL in PBS) were added (100 μL) to wells and incubated at 4 °C overnight. After washing with a 1-fold PBST solution (0.05% Tween 20), wells were blocked with 5% Blotting-Grade Blocker (Bio-Rad) diluted in 1-fold PBST for 2 h at room temperature. Incubation with analytes was carried out at room temperature for 2 h. After plate washing, 100 μL of (1 μg/mL) biotinylated mAbs were added to the wells, and the plate was re-incubated at room temperature for 2 h. Binding was detected using Streptavidin-HRP (R&D Systems Inc., Minneapolis, MN, USA) diluted 1:40 in 1-fold PBST (30 min. incubation at room temp.). The peroxidase reaction was performed using the TMB Peroxidase EIA Substrate Kit (Bio-Rad, Washington, DC, USA). Development was stopped by adding 50 μL of 2N H_2_SO_4_ to each well, and optical densities were measured at 450 nm (OD450) on a CLARIOstar Plus plate photometer (BMG LABTECH, Ortenberg, Germany).

### 2.8. Biological Activity Prediction

Due to the impossibility of performing a study of the biological activity of a large number of compounds in a short period, it was decided to conduct a preliminary assessment of the potential biological activity of compounds (**I**, **III**, **IV**, **X**, **XI**) using PASS-online modeling [22]. The drug-like properties of compounds (**I**, **III**, **IV**, **X**, **XI**) were determined by means of the SWISS-ADME web server.

## 3. Results

### 3.1. Cytotoxicity

As a result of the cytotoxicity study of resin acid derivatives, it was found that 12-sulfodehydroabietic acid (**I**) and its potassium salt (**IV**) showed weak toxicity and did not have a significant cytotoxic effect on the activity of the respiratory cells. It was not possible to calculate the IC50 correctly for these substances.

Based on the obtained data, for (**IV**), it was determined as >430.7 μg/mL; for (**I**)—as >517.9 μg/mL. The calculated ID50s for other test substances and graphs reflecting their cellular toxicity are shown in Figure 2 and Table 1.

It was found that potassium salts of abietic (**II**) and 12-bromodehydroabietic acids (**III**) had the highest cytotoxicity. For the rest of the compounds, the IC50 was above 400 μg/mL.

### 3.2. Antibacterial and Fungicidal Activity

The antibacterial and fungicidal activity of the studied compounds (**I**, **IV**, **VI**–**XI**) was screened in the concentration range from 1 to 50 mg/mL. The 1, 5, 10, 25, and 50 mg/mL doses of the solvent were studied.

The study results of the resin acids derivatives antibacterial activity are presented in Table 2 and Table 3, and the fungicidal activity in Table 4.

After analyzing the obtained data of antibacterial activity, it should be noted that direct dose-dependent activity was recorded almost for all studied compounds. A greater inhibitory effect was obtained with a concentration increase. Gum rosin (**VI**) showed great activity, and the studied compounds 12-sulfo-DAA (I) and 12-sulfo-DAA potassium salt (**IV**) did not show either antibacterial activity or fungicidal activity on any of the investigated microorganism strains.

Speaking about the growth inhibition selectivity of Gr+ and Gr− bacteria, it should be noted that the minimum inhibitory concentration against *Escherichia coli* was mg/mL: disproportionated rosin-5, gum rosin-5, tall oil rosin-5, abietic acid-5, abietinol-5, 12-bromo-DAA-1.

*Bacillus subtilis* cells, as representatives of gram-positive bacteria, are more sensitive to the action of the studied compounds. The minimum inhibitory concentration against *Bacillus subtilis* was (in mg/mL): disproportionated rosin-5, gum rosin-1, tall oil rosin-5, abietic acid-1, abietinol-1, 12-bromo-DAA-1.

Gum rosin (**VI**) showed the highest antibacterial activity, and its properties were recorded on the test cells of both Gr+ and Gr− bacteria.

### 3.3. Antiviral Activity

The assessment of the test compounds (**I–V**) antiviral activity by ELISA is shown in Figure 3.

The results (Figure 3) show that a slight decrease in the viral load was observed only upon stimulation of the cells **II** in the supernatants obtained from the cells, as compared with the control. The treatment of cells **II**, **III**, and **V** led to a decrease in the content of the viral protein NP inside the cells. The cell viability was additionally determined to assess a possible decrease in viral load due to the toxic effect caused by the treatment of the compound, at the used working concentrations of the test substances (Figure 4).

According to the obtained results, compounds **II**, **III** and, **V** led to a 50% decrease in viable cells. Compounds **IV** and **I** did not show antiviral activity and did not cause a significant decrease in cell viability. Most likely, the observed effect of reducing the viral protein inside the cell upon stimulation of them **II**, **III**, and **V** are due to the toxic effect of the compounds under study.

To clarify this fact, the dose-dependent effect was evaluated upon stimulation of infected A549 cells with compounds **II** and **IV** at doses of 20 and 200 μg/mL. As shown in the graphs in Figure 5, the compound **IV**, having low toxicity, did not significantly reduce the viral load at the considered concentrations. At the same time, a 10-fold decrease in the concentration of II from 200 μg/mL to 20 μg/mL led to a decrease in the antiviral effect.

It was shown that the compounds do not have antiviral activity against influenza A/California/07/09 H1N1pdm09 virus in vitro. The observed effect of reducing the viral load for compounds **II**, **III,** and **V** is presumably due to their toxic effect on A549 cells.

### 3.4. Biological Activity Prediction

The prediction of the biological activity spectrum of the studied compounds **I**, **III**, **IV**, **X**, **XI** (see Appendix A) revealed that compounds are determined as promising mucosal protectors. Compound **I** (80% probability) is capable of exhibiting antiviral activity (against influenza). The SWISS-ADME webserver did not detect any antiviral activity of compounds **I**, **III**, **IV**, **X**, **XI** (Appendix A). This is because the webserver does not give such a prediction but considers only the target proteins of animals and humans. However, there is an indirect indication of the toxicity of the compounds, for example, in compound (**IX**). The target for it can be enzymes responsible for the condensation of chromosomes, the separation of chromatids, and the removal of torsion stress during DNA transcription and replication. If these enzymes are blocked, the cell may die. The possibility of studying cytostatic activity requires a separate experiment. In addition, compound (**IX**) can be active against tumor necrosis factor (TNF). It is important to note that the SWISS-ADME web server does not indicate the nature of the effect of compounds on the receptors but notes its probability. That is, the compounds can both activate the receptor and block it. The results of testing compounds using the SWISS-ADME web server do not allow assessing the protective effect on mucous membranes.

## 4. Discussion

Many studies have been carried out to research the biological activity of resin acid derivatives, including cytotoxicity. The cytotoxicity of a series of dehydroabietic acid derivatives containing pyrimidine fragments was studied in [23]. Cytotoxicity against human liver cancer cells (HepG2), human breast cancer cells (MCF-7), human colon cancer cells (HCT-116), human lung cancer cells (A549), and normal human liver cells (LO2) was evaluated using MTT analysis in in vitro experiments. As a result, cytotoxic activity screening showed that most of the compounds exhibited moderate to high levels of cytotoxicity against these cancer cell lines.

The toxicity value is relative and is estimated concerning intact cells (not stimulated by any substances). Optical density values in intact cells reflecting the activity of NADPH-dependent cellular oxidoreductases are taken as 100%. This means a 100% survival rate. A decrease in this value for the tested compounds indicates their toxic effect.

The IC50 was calculated in our study. The concentration of the test substance at which the cell viability is reduced by up to 50%. It is impossible to determine the IC50 for some compounds because their solubility will be lower than the possible toxic concentration.

Resin acids affect a wide range of bacteria and fungi studied by electron microscopy [6]. Sodium salts of abietic acid protect against influenza and rabies viruses. Mixtures of sodium salts of abietic and other resin acids have bactericidal activity against Staphylococcus aureus [24].

It was found that abietic acid inhibits key features of bacterial growth, such as the ability to form colonies, the activity of adenosine triphosphate (both planktonic and biofilm), acid formation, and biofilm formation [25]. Abietic acid was identified as bacteriostatic. These substances caused minimal damage to the bacterial membrane.

The antibacterial activity of substances depends on the possibility of their entry into a cell or interaction with components of integumentary structures, causing a violation of their integrity. Escherichia coli is a representative of gram-negative bacteria, which lacks a multilayer cell wall; in other words, it is rather thin. The main structural element of the cell wall is peptidoglycan (murein). The murein network is several tens of layers in Gram-positive bacteria. The composition includes teichoic acids. Gram-negative bacteria contain only 1–2 layers of murein. Comparing integumentary structures, it should be noted that in addition to the cytoplasm’s inner membrane, an additional or outer membrane is located on top of the peptidoglycan layer. The thickness of this membrane exceeds the size of the peptidoglycan monolayer. The cytoplasmic membrane has a complex three-layer structure and is characterized by pronounced selective permeability. This can undoubtedly contribute to the features of the manifestation of the activities of the studied compounds on bacterial cells.

Not all studied derivatives of abietic acid exhibit fungicidal properties. *Candida tropicalis* showed the greatest resistance to compounds. It can be assumed that this is due to the structural features and component composition of the yeast cell wall.

Previously, we detected that the growth suppression of *Candida tropicalis* cells was found only in potassium salt of 12-bromo-DAA (**III**). The growth inhibition zone was 1.1 ± 0.24 and 1.9 ± 0.16 mm at concentrations of 25 and 50 mg/mL, respectively [26]. A significant antifungal effect was recorded for disproportionated rosin (**VIII**).

One study is currently underway that evaluates the antiviral activity of resin acids and their derivatives. In the paper [27], the authors evaluated the antiviral activity of abietic and dehydroabietic acid and controls [acyclovir (ACV) and heparin (HEP)] against HHV-1 and HHV-2 (herpes simplex viruses types 1 and 2) using the endpoint titration method (EPTT) with some modifications. It was found that methylabietate, abietinal, 8,13 (15)-abietadiene-18-oic acid, methylabieta-8,13 (15)-diene-18-oate, 8,13 (15)-abietadien-18-ol and dehydroabietinol acetate exhibited a significant broad spectrum of antiherpetic activity. However, they showed moderate activity against HHV-2 and were not active against HHV-1. It is impossible to establish a relationship between the structural activity and the hydroxyl group at C18 in a series of dehydroabietic derivatives against-HHV-2 activity. This was because the activities between the ester and hydroxyl groups were comparable.

The antiviral activity of various dehydroabietans against herpes simplex virus type 1 (HSV-1) was also determined by endpoint titration (EPTT) [28]. It was found that only dehydroabietinol acetate and dehydroabietinol benzoate reduced HSV-1 replication at concentrations below 100 mg/mL. Dehydroabietane, which showed the highest antiviral activity, turned out to be dehydroabietinol acetate.

In [29], a study of the antiviral activity of dehydroabiethylamine salts with modified organic acids against the influenza A/California/07/09 (H1N1) pdm09 strain was carried out. It was shown that the modification of dehydroabiethylamine with counterions of organic acids did not lead to an increase in antiviral activity. Let us assume that organic acid has the same activity. In that case, the combination of an acid with dehydroabiethylamine leads to a loss of antiviral activity (a decrease in the selectivity index) of the obtained salt.

## 5. Conclusions

Thus, the cytotoxicity and antibacterial, fungicidal, and antiviral activities of several compounds were studied. It was found that gum rosin (**VI**) showed great activity, and the studied compounds (**I**) and (**IV**) did not show either antibacterial activity or fungicidal activity on any of the studied microorganism strains. It was shown that a slight decrease in the viral load in the supernatants was observed upon stimulation of cells (**II**) compared with the control. The observed effect of the viral load reducing for compounds (**II**), (**III**), and (**V**) is presumably due to their toxic effect on A549 cells. Using PASS-online modeling the biological activity spectrum prediction showed that compound (**I**) could exhibit antiviral activity against the influenza virus.

These results indicate the potential of resin acid derivatives as a starting point for the extensive studies, both in vitro and in vivo, and reducing the toxicity of the resulting compounds, which are more effective from a biological point of view.

## Figures and Tables

**Figure 1 molecules-26-03836-f001:**
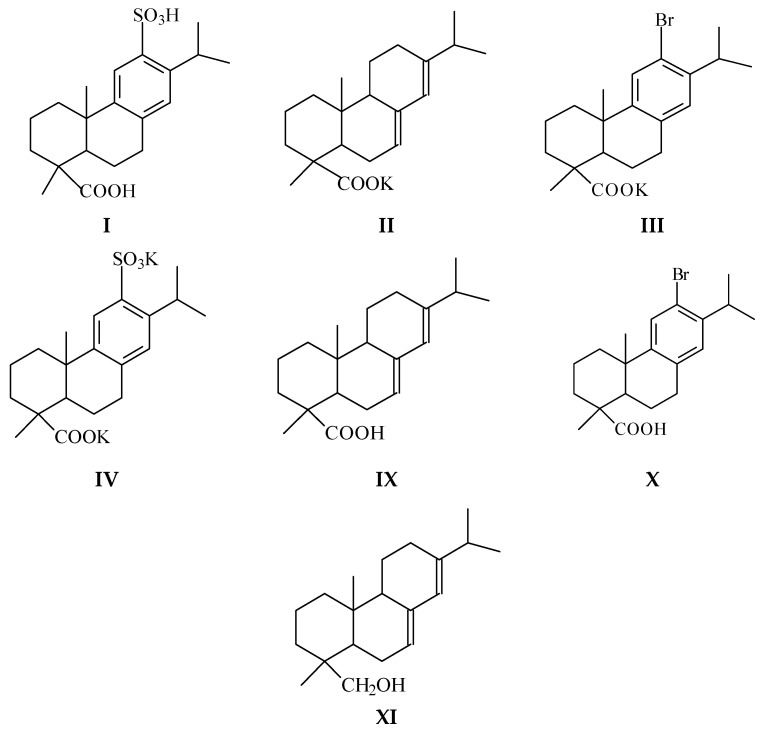
Compounds: (**I**). 12-sulfodehydroabietic acid, (**II**). Potassium abietate, (**III**). Potassium 12-bromodehydroabietate, (**IV**). Potassium 12-sulfodehydroabietate, (**IX**). Abietic acid, (**X**). 12-bromodehydroabietic acid, (**XI**). Abietinol.

**Figure 2 molecules-26-03836-f002:**
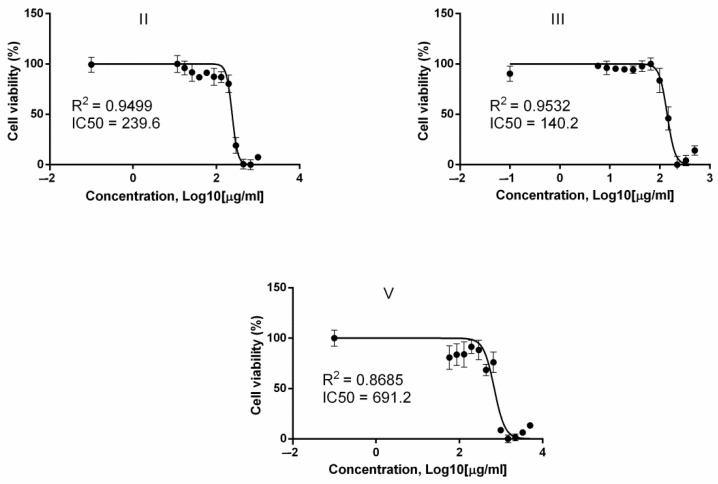
The assessment of the compounds **II**, **III,** and **V** cytotoxicity in cell culture A549. The cell incubation time with the test substances was 24 h—the method for the cytotoxicity assessment, the reduction of tetrazole (MTT assay).

**Figure 3 molecules-26-03836-f003:**
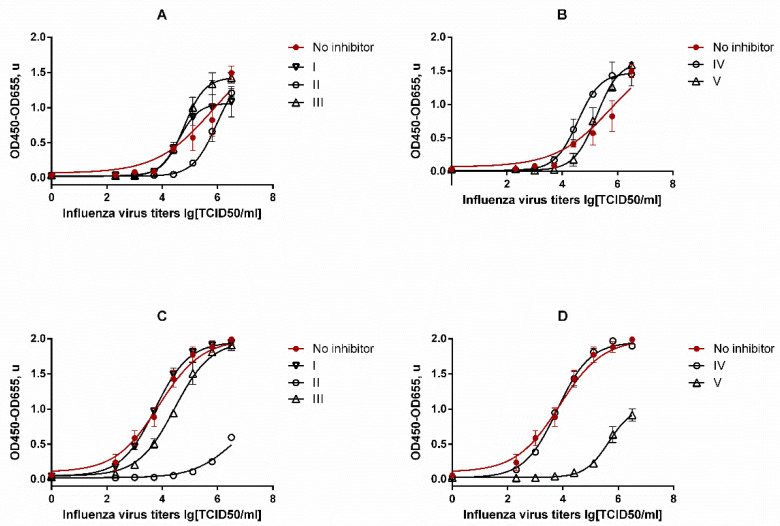
The assessment of the investigated compounds’ antiviral activity by ELISA. (**A**,**B**) the content of virions in supernatants obtained from cells upon stimulation with compounds **II**, **III**, and **V** (**A**) and **IV**, **I** (**B**). (**C**,**D**) The viral NP protein content inside the cell upon stimulation with **II**, **III**, and **V** (**C**) and **IV**, **I** (**D**) preparations. The ordinate scale (it is expressed in optical density units) is directly proportional to the content of the viral NP protein in the samples. The data obtained for HAV-infected cells which are not stimulated with substances are shown in red.

**Figure 4 molecules-26-03836-f004:**
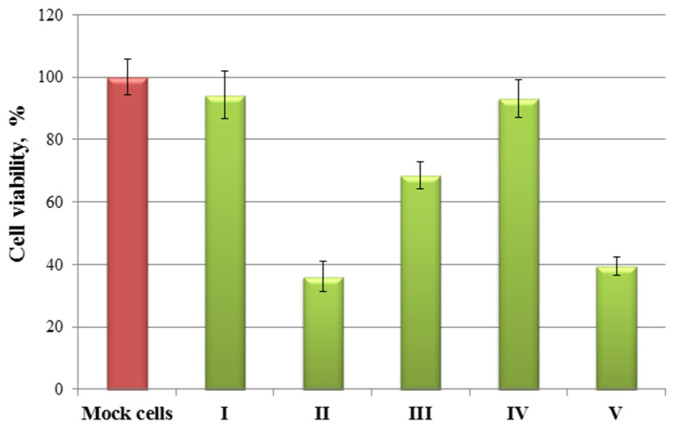
The cell viability when stimulated with compounds (**I**–**V**) at concentrations that were used to assess antiviral activity.

**Figure 5 molecules-26-03836-f005:**
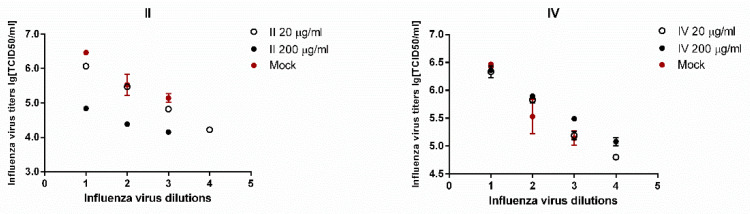
The dose-dependent antiviral effect obtained for compounds **II** and **IV**.

**Table 1 molecules-26-03836-t001:** Active doses of the studied compounds.

Compounds	Determined IC50, μg/mL	Working Concentration Used for Assessment of Antiviral Activity, μg/mL
**II**	239.6	200
**III**	140.2	50
**V**	691.2	50
**IV**	>430.7	200
**I**	>517.9	500

**Table 2 molecules-26-03836-t002:** Antibacterial properties compounds against *Escherichia coli*.

Compounds	The Zone of Growth Inhibition, mm				
	Compounds Concentration, mg/mL				
	1	5	10	25	50
**VIII**	0	1.0 ± 0.14	6.1 ± 0.02	6.1 ± 0.02	7.3 ± 0.02
**VI**	0	3.0 ± 0.10	15.0 ± 1.15	20.0 ± 2.50	-
**VII**	0	1.0 ± 0.10	2.0 ± 0.03	3.0 ± 0.08	Does not diff
**IX**	0	1.0 ± 0.10	1.0 ± 0.10	2.0 ± 0.24	3 ± 0.14
**XI**	0	1.0 ± 0.10	1.0 ± 0.10	3.5 ± 0.21	5 ± 0.14
**X**	0.9 ± 0.10	2.0 ± 0.07	7.6 ± 0.03	4.6 ± 0.05	Does not diff
**I**	0	0	0	0	0
**IV**	0	0	0	0	0

Note: 0—no zone; - was not investigated «does not diff»—the substance does not diffuse.

**Table 3 molecules-26-03836-t003:** Antibacterial properties compounds against *Bacillus subtilis*.

Compounds	The Zone of Growth Inhibition, mm				
	Compounds Concentration, mg/mL				
	1	5	10	25	50
**VIII**	0	1.0 ± 0.15	3.4 ± 0.06	6.2 ± 0.04	8.5 ± 0.04
**VI**	4.0 ± 0.15	8.0 ± 1.00	11.0 ± 2.20	12 ± 2.15	-
**VII**	0	1.0 ± 0.20	1.0 ± 0.15	2.0 ± 0.09	Does not diff
**IX**	1.0 ± 0.15	2.1 ± 0.03	2.0 ± 0.05	4.0 ± 0.10	5.0 ± 1.04
**XI**	1.0 ± 0.15	2.2 ± 0.05	2.0 ± 0.15	4.0 ± 0.12	4.0 ± 0.09
**X**	3.1 ± 0.25	9.5 ± 0.05	9.7 ± 0.04	10.5 ± 0.03	Does not diff
**I**	0	0	0	0	0
**IV**	0	0	0	0	0
DMSO	0				

Note: 0—no zone; - was not investigated.

**Table 4 molecules-26-03836-t004:** Fungicidal properties compounds against *Candida tropicalis*.

Compounds	The Zone of Growth Inhibition, mm				
	Compounds Concentration, mg/mL				
	1	5	10	25	50
**VIII**	0	3.0 ± 0.15	5.4 ± 0.06	6.8 ± 0.20	10.5 ± 1.25
**VI**	0	1.0 ± 0.10	1.5 ± 0.15	2.0 ± 0.09	3.0 ± 0.20
**VII**	0	0	0	1.0 ± 0.10	1.0 ± 0.20
**IX**	0	0	0	1.0 ± 0.11	1.0 ± 0.15
**XI**	0	1.0 ± 0.10	2.0 ± 0.10	4.0 ± 0.25	4.2 ± 0.15
**X**	0	0	0	1.1 ± 0.24	1.9 ± 0.16
**I**	0	0	0	0	0
**IV**	0	0	0	0	0
DMSO	0				

Note: 0—no zone; - was not investigated.

## Data Availability

Now new data were created or analyzed in this study. Data sharing is not applicable to this article.

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
