# Peer review of "Rosin Derivatives as a Platform for the Antiviral Drug Design"

_molecules, 2021, doi:10.3390/molecules26133836_

Round 1

Reviewer 1 Report

In this ms by Larisa Popova et al. the in vitro activity of rosin derivatives are presented. The experimental part is well-developed and the compounds have been completely characterized, I think, the manuscript is suitable for publication in Molecules. However two remarks need to be adressed: 
1. Please add the molecule structure of the titled compounds in the indroduction's main text. 

2. The manuscript is not using a QSAR modeling, instead a predictory activity using several databeses is employed. Thus, it must refer as that.

3. The drug-like properties should be added in section 3.4 as well. To adress that, a suitable web server such as SWISS-ADME can be employed.

Author Response

  1. Please add the molecule structure of the titled compounds in the introduction's main text.

The molecule structure of the compounds was added in the introduction's main text. All changes are reflected in the article.

  1. The manuscript is not using a QSAR modeling, instead a predictory activity using several databeses is employed. Thus, it must refer as that.

You're right. To predict the spectrum of biological activity of compounds, we used PASS-online modeling (Prediction of Activity Spectra for Substances). All changes are reflected in the article.

  1. The drug-like properties should be added in section 3.4 as well. To address that, a suitable web server such as SWISS-ADME can be employed.

In section 3.4. the results of a study of the drug-like properties of compounds obtained using the SWISS-ADME web server have been added. All changes are reflected in the article.

Reviewer 2 Report

In this paper, the cytotoxicity and antibacterial, fungicidal and antiviral activities of several compounds were studied. It was found that gum rosin (VI) showed great activity, and and shown that a slight decrease in the viral load in the supernatants was observed upon stimulation of cells (II) compared with the control. These results indicate the potential of resin acid derivatives as a starting point for extensive studies, both in vitro and in vivo, and solving the problem of reducing the toxicity of the resulting compounds, which are more effective from a biological point of view. These findings are important and helpful to understand activities of these series of compounds. However, the work was not very well presented. From these details, I think that this manuscript is appropriate for publication as a full paper in Molecules after major revisions and answers for the questions, suggestions and/or corrections.

  • Abstract: too cumbersome, please simplify.
  • Keywords must be in alphabetical order.
  • The compounds mentioned in the introduction can be placed in the text.
  • Synthesis of compounds and their identification: Give a subtitle to make the reader clearer, and the characterization method should be unified, for example 1H NMR, 13C NMR and HRMS.
  • The results and discussion section has too much work on previous people.

Author Response

1.Abstract: too cumbersome, please simplify.

Reduced the information in the abstract. All changes are reflected in the article.

2.Keywords must be in alphabetical order.

Keywords have been sorted alphabetically. All changes are reflected in the article.

3.The compounds mentioned in the introduction can be placed in the text.

The compounds indicated in the introduction have been placed in the text. All changes are reflected in the article.

4.Synthesis of compounds and their identification: Give a subtitle to make the reader clearer, and the characterization method should be unified, for example 1H NMR, 13C NMR and HRMS.

Subtitles were given for a clearer perception for readers. The method of characterization was also unified. All changes are reflected in the article.

5.The results and discussion section has too much work on previous people.

The discussion section assumes the interpretation and explanation of the meaning of the research results in the light of what has already been discussed in the existing literature on this issue. Therefore, we have presented in this section the existing research and the results of our research. But all the same, we took into account your comments and reduced some of the work of previous people. All changes are reflected in the article.

Round 2

Reviewer 2 Report

Acceptable after format check